# OpenReview forum: "Gradient-based Jailbreak Images for Multimodal Fusion Models"
_ICLR.cc/2025/Conference — Submitted to ICLR 2025_

### Official Review · Reviewer_nuLw · 2024-11-02

**Soundness:** 1
**Presentation:** 1
**Contribution:** 1
**Rating:** 3
**Confidence:** 4

**Summary:**

This paper proposes a gradient-based jailbreak method for multimodal fusion models. The authors introduce tokenizer shortcuts to solve the problem of continuous optimization not being carried out in the multimodal fusion model due to the discretization of input modalities. The experimental evaluation is carried out on the Chameleon multimodal fusion model. The results show that their method can trigger the generation of harmful information.

**Strengths:**

1. The approach is straightforward, relying on a fully connected network structure to approximate image tokenization.
2. The research addresses an important problem by targeting vulnerabilities in multimodal fusion models.

**Weaknesses:**

1. The effectiveness of the proposed approach is not well-validated. Without the tokenizer shortcut, the method's performance declines significantly, suggesting it may lack robustness in different settings.

2. From Table 2, the attack success rate drops when adding the refusal prefix part. The enhanced loss function, which aims to reduce the probability of generic refusal tokens, does not demonstrate a clear benefit in the experiments.

3. The approach's effectiveness is further limited when defenses are in place, raising concerns about its resilience against common protective measures.

4. Practical applicability is limited as the approach relies on assumptions that may not align with realistic conditions.

4.1 In direct attack scenarios, the method presumes the target model has been modified to include the shortcut, but it is unlikely defenders would incorporate this modification.

4.2 The approach also lacks sufficient transferability, reducing its usability across different models or settings.

5. The compared baselines are limited, just focusing primarily on text-based attacks GCG. A broader selection of attack methods would improve the robustness of the evaluation.

6. The use of $\Delta$PPL to measure adversarial prompt effectiveness lacks sufficient validation as a reliable metric.

**Questions:**

1. How can the robustness of the proposed method be improved to maintain effectiveness across diverse settings, especially in the absence of the tokenizer shortcut? Can the authors evaluate performance under different tokenization schemes?

2. Why do we need the enhanced loss function?

3. Can the authors evaluate their method on other defenses, such as those mentioned by Jain et al. [1]?

5. What are the ablation results of changing the number of layers in the fully connected network or replacing it with other simple architectures?

6. Can the authors include additional baseline methods to more comprehensively assess the robustness and effectiveness of the proposed method, such as the FGSM, PGD, or any other reliable attack methods?

7. How can $\Delta$PPL be further validated as a reliable metric? For example, evaluating $\Delta$PPL's effectiveness in multimodal fusion models with an F1 score would provide a clearer, more reliable assessment.

[1] Jain, N., Schwarzschild, A., Wen, Y., Somepalli, G., Kirchenbauer, J., Chiang, P. Y., ... & Goldstein, T. (2023). Baseline defenses for adversarial attacks against aligned language models. arXiv preprint arXiv:2309.00614.

---

> ### Author Response · Authors · 2024-11-14
>
> Thank you for your time and valuable feedback. We wrote a [general response](https://openreview.net/forum?id=wNg0LibmQt&noteId=0fWxmenJlb) with clarifications about our contributions and threat model. We believe some of the weakness raised may be motivated by a not-so-great explanation of the threat model and motivation in the paper. We are working to implement this together with other feedback from reviewers. We hope the general response and our comments below may clarify this, but we are happy to discuss further. We address the specific weaknesses below:
>
> > The effectiveness of the proposed approach is not well-validated.
>
> We discuss the effectiveness of the method in our specific white-box threat model (more on this in the general response). Additionally, we focus on presenting a method that enables optimization. We expect future work to look for novel optimization objectives that, combined with our optimization method, can result in more transferable attacks. Concurrent work suggests transferability is not a trivial problem to solve [1]
>
> [1] Schaeffer, Rylan, et al. "When Do Universal Image Jailbreaks Transfer Between Vision-Language Models?." arXiv preprint arXiv:2407.15211 (2024).
>
> > From Table 2, the attack success rate drops when adding the refusal prefix part.
>
> In general, optimizing text tokens is a very brittle process. As we add more objectives, it is easier to find solutions that do not necessarily suit the attacker objective. For example, we find that reducing the probability of “I am sorry” increases the probability of alternatives like “Unfortunately, as an AI assistant…” while not necessarily maximizing non-refusal. This is why we believe only optimizing for the target tokens is more effective.
>
> > The approach's effectiveness is further limited when defenses are in place
>
> We like to see this negative result as a positive and interesting finding by itself. There is no apparent reason why we should expect text safeguards to transfer to white-box image attacks. In fact, in the original Circuit Breaker work, they found that vision language models (adapter-based) should also be optimized on images to get an effective defense. In our case, we find that safeguards in early-fusion architectures may transfer better across modalities, this is worth exploring further by future work. As we are starting to see for Circuit Breakers in text, we expect that more complex optimization objectives will also find successful jailbreak images.
>
> > In direct attack scenarios, the method presumes the target model has been modified to include the shortcut, but it is unlikely defenders would incorporate this modification.
>
> We contextualize our work on a white-box threat model where the attacker has access to the weights of the model and can perform any manipulation. See the general response.
>
> > The approach also lacks sufficient transferability.
>
> Although we agree this is a negative result of our attack, we believe this is a very interesting finding that was also found by concurrent work in adapter-based architectures [1]. It is not trivial why text attacks like GCG transfer and tokenized images in our setup do not. This opens interesting questions for future work and finding transferable images should be a project of itself. Our focus was placed on building a method for gradient propagation that could enable future work to design better optimization objectives that could result in transferable images, among other improvements.
>
> > The compared baselines are limited, just focusing primarily on text-based attacks GCG. A broader selection of attack methods would improve the robustness of the evaluation.
>
> Our goal was never to find the best possible attack, but rather to introduce a new method that enables end-to-end optimization. The comparison with GCG tries to motivate why jailbreak images may be more efficient to optimize than text while obtaining similar performance. We believe that, the same way that we could use more complex baselines, our method could be used to instantiate more complex attacks that also increase the ASR. So we would like not to place to much focus on specific ASR.
>
> > The use of PPL to measure adversarial prompt effectiveness lacks sufficient validation as a reliable metric.
>
> Perplexity filtering has been widely studied as a filter for GCG prompts [2] and is presumably implemented in most production models like ChatGPT. Nevertheless, in a white-box threat model we do not expect attackers to deploy optional protections. We use PPL to illustrate an interesting property that may motivate future work: image attacks, unlike text, do not increase perplexity of the prompt. In fact, they find images that have high likelihood under the model.
>
> [2] Alon, Gabriel, and Michael Kamfonas. "Detecting language model attacks with perplexity." arXiv preprint arXiv:2308.14132(2023).

---

> > ### Author Response · Authors · 2024-11-14
> > **Answers to your questions**
> >
> > Now, we address your questions:
> >
> > > How can the robustness of the proposed method be improved to maintain effectiveness across diverse settings, especially in the absence of the tokenizer shortcut? Can the authors evaluate performance under different tokenization schemes?
> >
> > As we have seen with GCG on text prompts, it is likely that better optimization methods can achieve better transferability. As we mentioned above, this is still an open question that does not seem trivial to solve given existing work. Also, we only have Chameleon as an open-source early-fusion model. As more models are released, it is likely that ensembling over several models may increase transferability as in the original GCG work.
> >
> > We are not sure what you mean with “different tokenization schemes”. Could you clarify?
> >
> > > Why do we need the enhanced loss function?
> >
> > We ablated different natural choices for the loss function to illustrate performance and help future work on optimization objectives. We find that using longer contextualized targets for each prompt results in a huge increase from only optimizing for the token “Sure”.
> >
> >  > Can the authors evaluate their method on other defenses, such as those mentioned by Jain et al. [1]?
> >
> > Since we focus on a white-box threat model, we evaluate only on defenses that are directly baked into the model weights and cannot be trivially disabled by adversaries. The strongest available defense for white-box attacks is Circuit Breakers. Circuit Breakers aim to modify the weights in such a way that even with white-box access attackers cannot access the knowledge. All measures in the referenced paper assume black-box deployment of the model. Nevertheless, our perplexity analysis could be seen as the implementation of the most popular defense (self-perplexity filtering) in the cited work by Jain et al.
> >
> > > What are the ablation results of changing the number of layers in the fully connected network or replacing it with other simple architectures?
> >
> > We evaluated also a linear layer. Results are around 20% lower ASR for the best performing attacks. We will try to include an ablation analysis in the Appendix.
> >
> > > How can PPL be further validated as a reliable metric?
> >
> > This has been explored in previous work, as the one you referenced earlier by Jain et al. However, perplexity filtering would again be a defense on a black-box threat model and we include the results to illustrate the behavior of the attack (i.e. we find images with high likelihood under then model unlike text attacks).

---

> > > ### Comment · Reviewer_nuLw · 2024-11-27
> > >
> > > Thanks to the authors for their response to my feedback, which addressed some of my concerns. However, I still have reservations about a few key points in the article, especially in terms of method validity validation and benchmark selection.
> > >
> > > Regarding the authors' rationale for choosing to use a white-box threat model for evaluation, I understand that the purpose is to assess the performance of the model in a worst-case scenario. The idea of modifying the structure of the model to produce effective counter samples is indeed innovative, but testing the model with permission to modify it tends to make the paper's methodology much less actionable in practice. Although the success rate of jailbreaking may be higher in this setup, the general applicability of such results remains questionable in real-world application scenarios. Therefore, an assessment approach that relies solely on white-box attacks may not truly reflect the most representative abuse risk scenarios.
> > >
> > > In addition, the authors have some limitations in their choice of benchmarks, mainly focusing on the evaluation of textual attacks (e.g., GCGs), while comparisons of other types of attack methods are insufficient. Although the authors clearly state that their method is not aimed at finding the optimal attack strategy, the lack of comparison of more types of attack methods makes it difficult to comprehensively assess the generalizability and robustness of the proposed method. In order to validate the effectiveness of the proposed method more effectively, it is necessary to add more comparisons of multiple attack methods.
> > >
> > > In summary, although the authors have proposed some novel ideas, there are still some deficiencies in method validation, attack effectiveness and experimental design. The existing experimental results have not yet fully proved the effectiveness of their methods in practical applications, so this paper has not yet demonstrated sufficient contribution and practical application value.

---

> ### Author Response · Authors · 2024-11-27
>
> We thank the reviewer for taking the time to acknowledge our response and providing constructive feedback.
>
> We agree that applicability from the perspective of an adversary that wants to exploit a third-party system against their policies is limited. As we tried to clarify in our general response, this is not the goal of our paper. However, we would like to emphasise that our methods have wide applicability for the community as an adaptive evaluation tool for model and defense developers, who have full access to their systems. These two use cases ---attacking third-party systems versus worst-case evaluations--- should be understood separately.
>
> For understanding system vulnerabilities, it is often better to evaluate as many attack vectors as possible. As discussed in our paper, comparing different attack methods can be challenging and potentially less useful than considering them collectively during evaluations. Each method reveals different weaknesses, making them complementary rather than competitive. As we tried to motivate, our comparison with GCG serves as a contextualisation of this attack compared to equivalent optimisation on text space.

---

> ### Comment · Reviewer_nuLw · 2024-11-27
>
> This reviewer appreciates the author's response to the reviewer's feedback and the further clarifications provided. The author has clearly stated in their reply that the paper's focus is not on attacking third-party systems, but rather on serving as an adaptive evaluation tool for model and defense developers. This clarification is crucial for understanding the aim of the research.
>
> However, to further strengthen the paper's persuasiveness and practical value, the reviewer suggests that the authors delve deeper into the following areas:
>
> 1. Given that the paper presents the effectiveness of attacks specifically on models with "shortcuts," the reviewer recommends that the author provide a detailed explanation of the core significance of the method where "developers attack their own modified models to uncover vulnerabilities." It would be helpful to clarify why this approach is valuable for model developers, particularly in terms of its role in identifying vulnerabilities and enhancing model security and robustness.
>
> 2. Please explain in what situations developers might employ this method to evaluate and improve their models. If possible, the author could include one or more case studies to demonstrate the effectiveness of this approach in practice. This would help readers better understand its application and impact.
>
>
> These additions would further enhance the depth of the paper and make it more practical and informative for model developers and defense researchers.

---

> ### Author Response · Authors · 2024-11-27
>
> We thank the reviewer again for their engagement and constructive feedback. We can no longer update the PDF but we are happy to discuss possible ways to strengthen our message as requested.
>
> 1. Adaptive evaluations are a gold-standard to evaluate defenses against adversaries since image classification models [e.g. 1]. In the LLM literature, we often see protections evaluated against naive and black-box models that appear to be robust but are often bypassed by better adversaries. Doing white-box evaluations to begin with can help identify vulnerabilities before deploying protections. A prominent example is unlearning [e.g. 2,3] where white-box evaluations have shown that knowledge is not deleted from model weights, unlike initially thought. It is in these scenarios, where we assume full access, adaptive evaluations can provide a lot of value to better understand whether our attempts to remove information were successful.
>
> [1] Carlini, Nicholas, and David Wagner. "Adversarial examples are not easily detected: Bypassing ten detection methods." Proceedings of the 10th ACM workshop on artificial intelligence and security. 2017.
>
> [2] Lynch, Aengus, et al. "Eight methods to evaluate robust unlearning in llms." arXiv preprint arXiv:2402.16835 (2024).
>
> [3] Łucki, Jakub, et al. "An adversarial perspective on machine unlearning for ai safety." arXiv preprint arXiv:2409.18025 (2024).
>
> 2. Regarding its application in practice, our own work can be seen as an illustrative use case of our technique. The original Chameleon model was claimed to be safe. Using optimisation techniques like ours can demonstrate that the original safeguards are brittle and hazardous knowledge can be elicited from the model using image inputs.
>
> We hope these can help clarify our motivation. If so, we will update the paper accordingly for camera-ready if accepted. We are happy to discuss further if the reviewer thinks our framing can be further improved to inform better safeguards and evaluations.

---

> > ### Comment · Reviewer_nuLw · 2024-11-28
> >
> > The reviewer thanks the authors for their response to the review comments and for further elaborating on the background and application of the methodology. The authors' explanation demonstrates the potential of adaptive assessment as an evaluation tool.
> >
> > However, there are still some challenges regarding its practical applicability. Specifically, the current description does not adequately demonstrate the direct benefits this approach can offer during the actual process of model development and deployment, especially in the context of developers' daily work.
> >
> > The reviewer thanks the authors again for their efforts and hopes these suggestions will be helpful for the advancement of the authors' research.

---

> ### Author Response · Authors · 2024-11-28
>
> We thank the reviewer for their comment.
>
> Our previous response clarified how these tool can be used to evaluate models and safeguards adaptively. This is a research artifact and we do not think this should be understood "in the context of developers' daily work".
>
> We believe our clarifications and paper updates have addressed all concerns raised by the reviewer.

---

> > ### Author Response · Authors · 2024-12-04
> >
> > As the rebuttal period ends, we believe we have addressed all the reviewer’s concerns and kindly invite them to reassess the score if deemed appropriate.

---

### Official Review · Reviewer_sE6H · 2024-11-03

**Soundness:** 3
**Presentation:** 3
**Contribution:** 2
**Rating:** 3
**Confidence:** 4

**Summary:**

This paper proposed jailbreak attacks on multimodal fusion models by introducing a *differentiable* tokenizer shortcut. This allows for continuous optimization of adversarial images intended to bypass model safeguards. It evaluates the effectiveness of such attacks on Chameleon models, achieving a higher attack success rate than text-only jailbreaks. The results suggest that representation engineering defenses for text attacks could also adapt to adversarial image inputs.

**Strengths:**

- **Well-structured:** The paper is well-written and describes the proposed method clearly.
- **Introduced Differentiable Tokenizer:** This paper proposes using a two-layer neural network to make image tokenization in a multimodal fusion model feasible, enabling continuous optimization and revealing its threats to jailbreak.

**Weaknesses:**

- The proposed method of modifying the model architecture (replacing the original tokenizer) to elicit the jailbreak does not make much sense; also, the perturbed (attacked) images lack transferability. Given that a text-based attack is already feasible to pose such threats, I tend to buy the proposed method that applies the traditional method of generating adversarial perturbations to a multimodal fusion model. This method, however, is neither novel nor practically applicable to my understanding.

- Using adversarial images to elicit model jailbreak is also not novel; the paper lacks some discussion and comparison with existing works on VLLM [1].

[1] Visual Adversarial Examples Jailbreak Aligned Large Language Models (AAAI 2024)

**Questions:**

- Could you provide more insights into why the experimental results demonstrated a higher Attack Success Rate (ASR) using the embedding shortcut compared to the 1-hot shortcut?
- While it is understandable that jailbreak images optimized for Chameleon-7B might not transfer effectively to larger models, have you explored or observed whether jailbreak images optimized on larger models could be effectively transferred to smaller ones, such as from a Chameleon-30B to a Chameleon-7B model?

---

> ### Author Response · Authors · 2024-11-14
>
> Thank you for your time and valuable feedback. We wrote a [general response](https://openreview.net/forum?id=wNg0LibmQt&noteId=0fWxmenJlb) with clarifications about our contributions and threat model. We believe some of the weakness raised may be motivated by a not-so-great explanation of the threat model and motivation in the paper. We are working to implement this together with other feedback from reviewers. We hope the general response and our comments below may clarify this, but we are happy to discuss further. We address the specific weaknesses below:
>
> > The proposed method of modifying the model architecture (replacing the original tokenizer) to elicit the jailbreak does not make much sense
>
> Under a white-box threat model, the goal is to find any method that is able to elicit knowledge that was stored in the model weights. In the text domain we have seen methods like refusal suppression, included as a baseline, that also require modifying the architecture or activations at inference time. It is important to contextualise our contributions in the white-box threat model, we will try to incorporate these details in the paper. (More on this in the general response)
>
> > The perturbed (attacked) images lack transferability.
>
> Although we agree this is a negative result of our attack, we believe this is a very interesting finding that was also found by concurrent work in adapter-based architectures [1]. It is not trivial why text attacks like GCG transfer and tokenized images in our setup do not. This opens interesting questions for future work and finding transferable images should be a project of itself. Our focus was placed on building a method for gradient propagation that could enable future work to design better optimization objectives that could result in transferable images, among other improvements.
>
> > Text attacks already achieve this attack.
>
> In Section 5.2 we discuss the main differences between text and image attacks. On the one hand, image optimization is more stable leading to higher attack success rates and, more importantly, to a significant speed up in the optimization process (at least 3x). Augmenting the attack surface and the tools available for adversarial evaluation of models and protections is important.
>
> > Lack of comparison to existing works of VLLMs.
>
> We are sorry if this was not clear enough. Our paper is motivated by the difference that exists between VLLMs and early-fusion models. The models explored in previous work (we will include a citation to the mentioned work), are adapter-based. This means that there is already a differentiable path between the output and the image spaces. However, these architectures are very limited and the field is moving towards image tokenization. Our work thus focuses on introducing a method that enables end-to-end gradient propagation (similar to what we saw in previous work with VLLMs) on this new architectures. This is discussed in the second paragraph of the introduction and the “Jailbreaking language models” in the preliminaries. Please, let us know if this is not clear enough so that we can clarify it in a future version of the paper.
>
>
> We now answer your questions:
>
> > Transferability in other setups.
>
> Images do not transfer from 30B to 7B either.
>
> > Why embedding attack has higher ASR than 1-hot?
>
> This is a great question and also related to a question by reviewer `zzub`. We will include a discussion comparing both methods in the future version of the paper. The embedding space gives a more expressive representation to find successful attacks since optimization is unrestricted. Optimizing the 1-hot representation encourages changes in the image that result in changes in the tokenization. Since the mapping from tokens to embeddings is deterministic and sparse, this can be seen as a regularization that may hinder the attack at the cost of better transferability if the shortcut is turned off.
>
> [1] Schaeffer, Rylan, et al. "When Do Universal Image Jailbreaks Transfer Between Vision-Language Models?." arXiv preprint arXiv:2407.15211 (2024).

---

### Official Review · Reviewer_Djns · 2024-11-03

**Soundness:** 2
**Presentation:** 3
**Contribution:** 1
**Rating:** 3
**Confidence:** 3

**Summary:**

This paper proposes a method generating jail-break images that cause early-fusion VL models (especially those with discrete image tokens) to generate harmful content when jail-break images are appended with harmful prompts.
Unlike adapter-based VL models, which do not use image token discretization, the discrete tokenization of images in early-fusion VL models makes direct optimization through gradients challenging and limits the applicability of existing methods.
To address this, the paper proposes a tokenizer shortcut that bypasses the discrete tokenization process by replacing quantization with a shallow MLP module, enabling the computation of gradients.
The experiments demonstrate the effectiveness of the proposed method for generating jail-break images under certain settings—specifically, a white-box attack.

**Strengths:**

1. This paper addresses an important reasearch topic of jail-breaking in VL-LLM models, considering the significant growing use of VL models in real world applications. Research in this direction seems essential.
2. This paper is well presented, making the paper easy to follow and understand.

**Weaknesses:**

1. There is a lack comparison or discussion with other condidates to make quantizqation differentiable. If the proposed method achieves very strong performance in generating jail-breaking iamges, current approach would be acceptable. However, it seems that the proposed method can generate jail-break images in very limited settings: with shortcut or non-transfer setting.
2. As far as i understand, the white-box attack scenario is important because, although it may be impractical and unrealistic, it serves as a useful benchmark for black-box attacks. However, for the "with shortcut" results, it effectively becomes equivalent to altering the model itself, which makes discussions of attack performance somewhat meaningless. Nonetheless, the proposed method is primarily evaluated using the shortcut when demonstrating its strong performance, (Table 1, 2, 3, 4).
3. Optimizing within the input (image and text) space is important, as it is a prerequisite for black-box settings or model transfer. However, as shown in Table 5, the proposed method fails to produce transferable samples and underperforms compared to the baseline.
4. (Minor) The paper seems to contain overclaims or insufficient explanations. For example:
	- The title of Table 3 is "Image jailbreaks outperform text attacks," but the proposed method performs worse than the text-only attack, GCG, in the Circuit Breaker setting. Additionally, comparing GCG with the proposed method "with shortcut" seems unfair, as "with shortcut" is equivalent to changing the model.
	- In discussions and future works, the paper states, "(412) Our work is the first attempt to jailbreak multimodal architectures using end-to-end gradient attacks" and "(423) this problem also persists in multimodal models,". I guess the "fusion-based model" shall be more appropriate.

**Questions:**

1. Please address the weaknesses.
2. (Suggestion) Moving the related work section front or refering that more related work is in the later part shall improve the understanding of the paper.

---

> ### Author Response · Authors · 2024-11-14
>
> Thank you for your time and valuable feedback. We wrote a [general response](https://openreview.net/forum?id=wNg0LibmQt&noteId=0fWxmenJlb) with clarifications about our contributions and threat model. We believe some of the weakness raised may be motivated by a not-so-great explanation of the threat model and motivation in the paper. We hope the general response and our comments below may clarify this, but we are happy to discuss further. We address the specific weaknesses below:
>
> > Comparison to other methods to make quantization differentiable.
>
> We found our shortcuts to be the most effective among several approaches we considered. Do you have any other suggestions we could evaluate against? The main contribution of our work is introducing a method that enables end-to-end optimization of images in new early-fusion architectures. We expect future work to combine this method with improved optimization objectives that can improve upon different axis of success like transferability.
>
> >  It seems that the proposed method can generate jail-break images in very limited settings
>
> Our method is meant to elicit dangerous capabilities in a white-box threat model with full access to the model. We explain why this is important and different applications (e.g. defense evaluation) in the general response.
>
> > Importance of white-box threat model.
>
> We agree with your point here. We will extend a bit on the importance of white-box threat models. As open-source models become more powerful, it is more likely that adversaries use these models to automate harmful activities rather than actively jailbreaking closed-source models. Uncovering vulnerabilities under white-box access is important to motivate the need of new defenses and serve as hard evaluations for such defenses.
>
> Additionally, white-box evaluations allow us to assess worst-case performance of models. For example, these methods can be used to stress test the models and evaluate whether they indeed possess harmful knowledge. This can be used to robustly evaluate defenses and unlearning methods.
>
> > Lack of transferability
>
> Although we agree this is a negative result of our attack, we believe this is a very interesting finding that was also found by concurrent work in adapter-based architectures [1]. It is not trivial why text attacks like GCG transfer across models and tokenized images in our setup do not. This opens interesting questions for future work and finding transferable images should be a project of itself. Our focus was placed on building a method for gradient propagation that could enable future work to design better optimization objectives that could result in transferable images, among other improvements.
>
> > Insufficient explanations.
>
> Thanks for bringing these two issues up. For the “image jailbreaks are superior” we were focusing on models without additional defenses. Also, comparison of Circuit Breakers is tricky since some of the successes for GCG are not really useful to solve the task and our result may slightly overestimate the success of the method and the differences are not that clear in that setup.
>
> On the second point, we definitely agree we should include early-fusion in the statement to make it more accurate. Thanks for the catch.
>
> Finally, to your question:
>
> > Suggestion to move related work
>
> We believe most relevant aspects of the related work are contained in the preliminaries section. What content did you find important and not self-contained in previous sections? This can help us assess whether it would be better to move the entire section or just include additional details in existing sections.

---

> > ### Comment · Reviewer_Djns · 2024-11-24
> > **Thank you for the authors' response**
> >
> > Thank you for the authors' response.
> > Unfortunately, I still have concerns regarding the setting of this work.
> >
> > As I understand, the author explained that the white-box threat model is important for two reasons:
> > 1. The importance of using open-source models to automate harmful activities.
> > 2. To assess the worst-case performance of the model.
> >
> > Regarding the first reason, I agree that research on whether open-source models can be misused for harmful activities is important. However, if we exclude considering the black-box transferability, it is unclear to me why adversarial samples and differentiable relaxation for quantization are suitable for this goal of making open-source models harmful.
> >
> > As far as I know, many attacks focus on optimizing samples because they can be utilized without altering the model and may transfer to other models. However, as in the case of this paper, where modifications and training of the model's pipeline or weights are both allowed, focusing optimization on adversarial samples for white-box attacks seems less efficient compared to other potential approaches. For instance, in practical scenarios, PEFT (Parameter-Efficient Fine-Tuning) is often used to adapt open-source models for specific tasks. One possible direction could be to apply PEFT to induce harmful responses.
> >
> > This seems problematic from the point of view of its use for worst-case performance in white-box settings, since there are more general candidates that could serve as baselines for evaluating whether open-source models can be misused. On the other hand, when considering worst-case results for black-box attacks, the need to modify the model inference pipeline (i.e., with shortcuts) suggests that the results may not represent a tight worst-case performance.
> >
> > Additionally, regarding the lack of transferability, the finding that creating transferable samples via images is challenging for early-fusion VL models is convincing, in that it aligns with concurrent work. However, it doesn't seem to me that the results are surprising or represent a significant contribution.

---

> > > ### Author Response · Authors · 2024-11-25
> > >
> > > We thank the reviewer for engaging in the rebuttal. We would like to briefly comment on the points raised in the response
> > >
> > > 1. There are other methods to elicit harmful behavior in white-box models.
> > >
> > > This is true and we never meant to suggest that our method is the best for all situations. Our paper presents a new attack vector for novel architectures and provides evidence that image inputs expand the attack surface and enable more efficient gradient-based attacks than text inputs.
> > >
> > > Our jailbreak images transfer across prompts, as we show in Table 4. This means that one could optimise an image on 10 inputs and obtain generations to any prompt. The cost of this attack would be similar to fine-tuning.
> > >
> > > 2. White-box optimisation may not represent a tight worst-case performance
> > >
> > > The goal of eliciting dangerous capabilities is not to tightly bound worst-case performance on black-box scenarios, but rather demonstrating the **existence** of worst-case dangerous behaviours (e.g. can the model provide detailed instructions to build a bomb?) The existence of this knowledge may condition deployment safeguards.
> > >
> > > 3. The results are not surprising
> > >
> > > Our paper is the first to demonstrate how to optimise end-to-end attacks on early-fusion models, and provides the community with a new method to red-team this novel architecture.

---

### Official Review · Reviewer_PXgY · 2024-11-04

**Soundness:** 3
**Presentation:** 3
**Contribution:** 3
**Rating:** 8
**Confidence:** 3

**Summary:**

This paper studies white box attacks against multimodal fusion models. This setting is considered interesting because these models convert all inputs - both text and images - into a shared tokenized space. This approach could make the models vulnerable to more efficient attacks through image optimization, since images offer a continuous space to optimize (unlike text which is discrete). In order to optimize potential attack inputs, they develop the tokenization shortcut method, mapping image embeddings to a continuous model input space before quantization. They find that for whitebox optimization attacks, images are more effective than text, however they do not beat other competitive baselines like representation engineering.

**Strengths:**

- The choice of studying robustness of multimodal fusion models is timely.
- The selection of research questions is fitting for a first study in a fast-paced field. The hypothesis that it may be easier to attack models with this architecture is interesting, and is very useful to study early in the uptake of architectures.
- The paragraph writing style is easy to read, and the work can serve as an interesting log of experiments for other practitioners.

**Weaknesses:**

- The choice of the two shortcut is not clearly explained in section 3. It would be useful to spell it out.
- It would be useful to have more qualitative analysis or at least examples of jailbreaking images vs images that fail.

**Questions:**

- Your experiments seem interesting, and it seems like you may have opinions on future work. While you have already provided motivation for experiment design, it would be useful to add more detail to your results so that it is easier to judge what puzzles are worth investigating. For instance, it would be great to spell out the details of transferability experiments. The observation itself is cool, but the current presentation of the work makes it so that readers will have to reimplement your work to get started on forming hypotheses.

---

> ### Author Response · Authors · 2024-11-14
>
> Thank you for your time and valuable feedback. We wrote a [general response](https://openreview.net/forum?id=wNg0LibmQt&noteId=0fWxmenJlb) with clarifications about our contributions and threat model. We address the specific weaknesses below:
>
> > The choice of the two shortcut is not clearly explained in section 3.
>
> The embedding and 1-hot space are the only two continuous input spaces to the model that allow us to create a fully differentiable end-to-end path. Is this type of motivation for the choice what you had in mind?
>
> > More qualitative analysis.
>
> We are working to include successful jailbreak generations from the models in the Appendix of the paper. With respect to images, most images just include perturbations that look pretty arbitrary to humans; we believe there are no observable differences between successful and unsuccessful images. We are happy to include several examples in the Appendix if you think this could improve the understanding of our method.
>
> > Ideas for future work.
>
> Thanks for bringing this up. Our work indeed was meant to introduce a method for gradient propagation and not necessarily the best attack possible. We believe there is a lot of room for future work to use this method in combination with better optimization strategies to obtain stronger attacks. Since there are no other open-source early-fusion models makes research also challenging at the moment because methods can easily overfit to the Chameleon architecture and not transfer more generally.  We have included several directions that naturally follow from our work in the discussion section. All our code is open-source so reproduction of our results should be easy for people to build upon our work. What details of transferability experiments would you like to see in the paper? We are happy to include those.

---

> > ### Comment · Reviewer_PXgY · 2024-11-19
> >
> > > The embedding and 1-hot space are the only two continuous input spaces to the model that allow us to create a fully differentiable end-to-end path. Is this type of motivation for the choice what you had in mind?
> >
> > Yes, this sort of spelling out would be great.
> >
> > > We are working to include successful jailbreak generations from the models in the Appendix of the paper.
> >
> > Cool! Is there any analysis of the ways in which the model is jailbroken? This is useful to provide for replications that may put more effort into making particular attack types work.
> >
> > > What details of transferability experiments would you like to see in the paper?
> >
> > I would love more details of your dataset, for transferability, and in other sections of the paper. It would really help contextualize your reported numbers and make it clear what experiments have not been attempted yet.

---

> > > ### Author Response · Authors · 2024-11-20
> > >
> > > We have uploaded an updated version of the paper with changes highlighted in blue. We have included:
> > >
> > > * Clarification on why we chose those input spaces.
> > > * Examples of successful jailbreaks in Appendix C
> > > * Details on the dataset. We have clarified that each prompt in JailbreakBench is a different task (i.e. there are no two prompts about the same goal, say building a bomb). This highlights that our transferability experiments demonstrate that this method can unlock different behaviours.
> > >
> > > Thank you again for your feedback, we believe these improvements will make our contribution clearer for all readers.

---

> > > > ### Comment · Reviewer_PXgY · 2024-11-27
> > > >
> > > > Thanks for adding these sections and clarifications. I will raise my score, and maintain support for this paper.

---

### Official Review · Reviewer_zzub · 2024-11-04

**Soundness:** 3
**Presentation:** 4
**Contribution:** 4
**Rating:** 8
**Confidence:** 4

**Summary:**

* The paper develops a white box gradient-based image jailbreak method for multimodal fusion models. Prior work on gradient-based image jailbreaks has focused on VLMs due to the lack of open source fusion models, but this has recently changed with the release of Chameleon.
* The core challenge of doing this with fusion models is that gradients do not flow through to the input image due to a non-differentiable step in tokenization.
* The authors solve this problem by introducing a novel “tokenizer shortcut” technique, where they train a small MLP to approximate the image tokenizer in a differentiable way. The tokenizer is then replaced by this approximation during adversarial image optimization, allowing gradient-based optimization to succeed.
* Two versions of the tokenizer shortcut are developed, one mapping directly to embedding space and one producing a one-hot vocabulary encoding.
* A comprehensive set of experiments are conducted. Key findings:
  * Both shortcut methods produce images with high Attack Success Rate, but only the 1-hot shortcut images transfer to versions of the model that do not use the shortcut.
  * Circuit breakers substantially reduce ASR
  * Jailbreak images transfer easily across prompts but do not transfer across models
* The authors also conduct a series of ablations, including on response prefix, softmax temperature, and number of train prompts.

**Strengths:**

* This is a novel method that solves the core challenge of creating gradient-based image jailbreaks for multimodal fusion models.
* Understanding the vulnerabilities in multimodal models is important for developing more robust systems, and gradient-based jailbreaking of fusion-based models has been under-explored.
* The authors use good baselines for their experiments (GCG and refusal direction attacks), and convincingly demonstrate the success of their method
* The experiments are thorough and informative, testing attack transfer as well as defence using circuit breakers. Several ablations are also performed.

**Weaknesses:**

* The dataset used is quite small, with only 80 prompts in the test set for direct attacks and 20 in the test set for transfer attacks. The results would be more convincing if done on a larger dataset. In addition, only a single dataset is tested.
* The paper does not include any examples of jailbroken model responses - these are helpful for qualitative understanding of the attack.
* With the exception of table 1, the results given are all for models using the tokenizer shortcut. It would be helpful to also include  the results when using the 1-hot jailbreak images on models without the shortcut in Tables 2 and 4.

**Questions:**

* Do the authors have an explanation for why the embedding space shortcut attacks do not transfer to non-shortcut models while the 1-hot attacks do?

---

> ### Author Response · Authors · 2024-11-14
>
> Thank you for your time and valuable feedback. We wrote a [general response](https://openreview.net/forum?id=wNg0LibmQt&noteId=0fWxmenJlb) with clarification about our contributions and threat model. We address the specific weaknesses below:
>
> > The dataset is quite small
>
> We agree with this, but decided to stick to what we consider to be the best jailbreak benchmark as of today. This benchmark already curated prompts from other existing benchmarks. Although we could extend results to other benchmarks, most prompts will be redundant and similar to those in JailbreakBench. We are happy to discuss further experiments if you think this can bring fundamental new insights to the method.
>
> > The paper does not include any examples of jailbroken model responses
>
> This is a great point! We are working to include those.
>
> > Include results without shortcut in Tables 2 and 4.
>
> Thank you! We are working to include these in the Appendix. We initially left them out since we are assuming a white-box threat model where the attacker can use the strongest attack available. See our general response for clarifications on the threat model.
>
> Now, we answer your question:
>
> > Do the authors have an explanation for why the embedding space shortcut attacks do not transfer to non-shortcut models while the 1-hot attacks do?
>
> Our intuition is that using the 1-hot attack creates an inductive bias where the gradients incentivize changes in the image that produce changes in the actual tokenization. Thus, the resulting image is more likely to be tokenized in a way that generalizes when the shortcut is turned off. However, using the embedding is likely to result in changes that do not necessarily affect tokenization and thus are lost once the shortcut is turned off. We will include this hypothesis in the discussion section.

---

### Author Response · Authors · 2024-11-14
**Clarifications about threat model and research goals**

We thank all the reviewers for their time and feedback. In this general response, we would like to clarify some aspects of the paper, especially regarding the threat model we operate in and why we believe this is relevant. We will try to clarify these points in the paper as well.

**Main goal of the paper**: design a new method that enables end-to-end optimization for jailbreak images in novel early-fusion architectures. This attack can be use to **elicit harmful information stored in models**. We believe this is a contribution in itself and future work can improve optimization targets to elicit more transferable attacks, as we have seen the community iterate on text attacks like GCG.

**Threat models and assumptions**: In our work, we operate under a *white-box threat model*. What this means is that the attacker can download the weights of the model and run it locally with any desired modifications and disabling any optional safeguards. This threat model is relevant for two reasons:

* As open-source models become more capable, attackers are likely to use open-source models for harmful tasks, rather than jailbreaking closed source models.
* White-box evaluations are important to understand worst-case behavior of models. Namely, if you are developing a model, white-box attacks can *elicit the worst-case performance of the model*, which you can assume to be an upper bound on black-box vulnerabilities. *White-box jailbreaks are thus also a useful tool to evaluate defenses.*

Since we operate in white-box threat model, we only consider the Circuit Breakers as a defense, as it is the strongest jailbreak defense available that is **directly baked into the model weights** and thus cannot be trivially disabled by adversaries (like additional filters on the inputs and outputs).

We include transferability results since we believe it is interesting to highlight that our findings agree with those of concurrent work that report transferability of image jailbreaks to be harder than expected and an open problem \[1\]. We expect future work to be able to use our shortcut technique to optimize other objectives that might result in more transferable images.

\[1\] Schaeffer, Rylan, et al. "When Do Universal Image Jailbreaks Transfer Between Vision-Language Models?." *arXiv preprint arXiv:2407.15211* (2024).

---

### Meta-Review · Area_Chair_J8hF · 2024-12-10

**Metareview:**

This paper presents a method for white-box gradient-based image jailbreak attacks on multimodal fusion models using a "tokenizer shortcut" for continuous optimization. The proposed method achieves higher attack success rates than text-based jailbreaks and is more computationally efficient.  A major limitation of the model is the white-box access assumed, which is not true for the majority of the strongest LLMs at the moment. In addition, the paper could benefit from stronger comparisons, including additional defense methods or transferability as mentioned by reviewers. Lastly, the comments of reviewers (e.g. overclaiming the comparison of image to text jailbreaking) should be amended in the paper before it's ready for publication.

**Additional Comments On Reviewer Discussion:**

Various experiments were requested by reviewers, including "Include results without shortcut in Tables 2 and 4", but the responses mention those will be added in the future. Therefore, I do not think the reviewers are convinced from the responses. In addition, some of the assumptions are pretty strong and the results do not necessarily justify those.

---

### Decision · Program_Chairs · 2025-01-22

Reject